# Impacts of Temperature and Time on Direct Nitridation of Aluminium Powders for Preparation of AlN Reinforcement

**DOI:** 10.3390/ma16041583

**Published:** 2023-02-14

**Authors:** Samuel Rogers, Matthew Dargusch, Damon Kent

**Affiliations:** 1School of Mechanical and Mining Engineering, The University of Queensland, Brisbane 4072, Australia; 2School of Science, Technology and Engineering, University of the Sunshine Coast, Maroochydore 4558, Australia

**Keywords:** powder metallurgy, aluminium nitride, metal matrix composite, Al/AlN

## Abstract

Aluminium nitride (AlN) is an important technical ceramic with outstanding strength and thermal conductivity that has important applications for advanced heat sink materials and as a reinforcement for metal-based composites. In this study, we report a novel, straightforward and low-cost method to prepare AlN powder using a vacuum tube furnace for the direct nitridation of loose aluminium powder at low temperatures (down to 500 ${}^{\circ }$C) under flowing high-purity nitrogen. Small amounts of magnesium powder (1 wt.%), combined with aluminium, promote nitridation. Here, we characterise the effects of time (up to 12 h) and temperature (490 to 560 ${}^{\circ }$C) on nitridation with the aim to establish an effective regimen for the controlled synthesis of an aluminium nitride reinforcement powder for the production of metal matrix composites. The extent of nitridation and the morphology of the reaction products were assessed using scanning electron microscopy and X-ray diffraction analyses. AlN was detected for all nitriding temperatures ≥ 500 ${}^{\circ }$C, with the highest yields of 80% to 85% obtained at 530 ${}^{\circ }$C for times ≥ 1 h. At this temperature, nitridation proceeded rapidly, and there was extensive agglomeration of the reaction products making it difficult to reprocess into powder. At lower temperatures around 510 ${}^{\circ }$C, a relatively high proportion of AlN was attained (>73% after 6 h) while retaining excellent friability so that it could be manually reprocessed to powder. The synthesised reinforcement consisted of micro- or nano-crystalline AlN comingled with metallic aluminium. The ratio of AlN and metallic aluminium can be readily controlled by varying the nitriding temperature. This provides a flexible and accessible method for the production of AlN-reinforcement powders suited to the production of metal matrix composites.

## 1. Introduction

Aluminium nitride (AlN) is a preferred ceramic material for a range of technical applications due to an excellent combination of physical and functional properties. These include high corrosion resistance, high hardness, an excellent strength-to-weight ratio, good bio-compatibility, a low thermal expansion coefficient and high thermal conductivity [1,2,3,4,5]. This combination of properties makes AlN a desirable choice for applications such as heat sinks [6,7], semiconductors [8], coatings for optical devices [9], surface hardening, wear coatings and as a reinforcement for metal-based composites [10,11].

Commercial synthesis of AlN is conducted through a variety of technologies, including plasma nitriding [12,13], ball milling and annealing [14], evaporation [15], carbothermal reduction [16] and high-temperature direct gas nitriding [17,18]. These methods typically require specialist equipment and involve substantial energy consumption. In contrast to conventional methods for the synthesis of AlN, direct nitriding of aluminium under a nitrogen atmosphere at low-temperature (i.e., furnace temperatures below the melting point of Al at 660 ${}^{\circ }$C) offers a simple, straightforward and economic preparation method [19,20].

Direct nitridation of aluminium is inhibited by the thermodynamically stable oxide layer on its exposed surface as well as its high chemical affinity to oxygen [5,21]. Theoretically, extremely low oxygen partial pressures of around $10^{-50}$ Pa at 600 ${}^{\circ }$C are required to reduce the surface oxide. However, such low pressures are not practically attainable [22]. In this respect, careful control of the atmosphere to exclude oxygen is necessary for nitriding to take place. Furthermore, elements, such as magnesium or lithium, may be employed to getter oxygen from the furnace environment and disrupt the oxide layer to initiate the formation of AlN [23,24].

Lumley et al. [25] showed that, at low temperatures (400 to 520 ${}^{\circ }$C), trace additions of Mg suppress alumina and, in turn, promote the growth of MgO (magnesium oxide) and MgAl${}_{2}$O${}_{4}$ (magnesium aluminate) at the exposed surface. The Mg, which is present as a vapour at the nitriding temperatures, thus, disrupts the surface oxide and exposes the underlying metallic Al to the nitrogen atmosphere so that AlN can form [24,26]. Research has shown that, for Al powders, the optimal amount of Mg to promote nitriding is typically between 0.1 wt.% and 1.0 wt.% [27].

Various processes have been developed that utilise the low-temperature direct nitridation of Al powder, incorporating small amounts of Mg for the preparation of in situ reinforced Al/AlN metal matrix composites (MMCs) [11,28]. The AlN forms as layers on Al surfaces exposed to the nitrogen atmosphere. The nitride layers are comprised of fine poly-crystalline AlN intermingled with small amounts of metallic Al, which form percolating networks on the exposed Al powder surfaces. In situ methods for the preparation of Al/AlN MMCs offer significant advantages over ex situ methods, including the ability to precisely control the morphology and size of the AlN crystallites, which can be varied from nano- to micrometer length scales by adjusting the nitriding conditions.

The in situ formed AlN possesses greater thermodynamic stability, fewer microstructure defects and enhanced interfacial bonding with the matrix [11,20]. Powder metallurgy-based methods for consolidation of ex situ reinforced Al/AlN MMCs using conventional commercial AlN powders are compromised by the inert nature of the reinforcement at the low temperatures required for sintering Al. The high melting temperature ceramic does not actively participate in diffusion dominated sintering processes leading to restricted densification and poor interfacial bonding between reinforcement and matrix. This has led to considerable focus on in situ methods for the fabrication of Al/AlN MMCs.

Despite this, there are potential advantages in decoupling the process for nitridation from that for consolidation of the MMC as it can potentially facilitate greater microstructure control, including over the form, proportion and distributions of the reinforcement phase. Additionally, excessive in situ nitride formation during powder metallurgy preparation of MMCs may act as a rigid skeleton that inhibits densification so that there is a practical limit to the amount of in situ AlN reinforcement, which can be accommodated for standard press and sinter methods, beyond which, thermo-mechanical processing is required for consolidation.

The motivation for this research was to develop a method for the production of an AlN-reinforcement powder suitable for powder metallurgy-based production of MMCs. We anticipate that the reinforcement can possess similar physical characteristics to in situ formed AlN, which imparts outstanding strength and wear resistance to MMCs. Furthermore, the close intermingling of metallic Al with AlN can potentially enhance sintering and bonding with reinforcement for powder metallurgy-based preparation of MMCs.

The research to date has primarily focused on the preparation of in situ reinforced Al/AlN composites involving relatively low AlN phase proportions, and there have been more limited investigations into the low-temperature (below 600 ${}^{\circ }$C) nitridation of Al powders to obtain high AlN phase fractions [3,4,10,11,19,20,21,23,29,30]. Here, we systematically investigate the impacts of time and temperature on the nitridation of loose Al powder containing 1 wt.% Mg to establish an effective regimen for the controlled synthesis of an AlN reinforcement. Scanning electron microscopy and X-ray diffraction analyses are employed to investigate nitridation, including the phase fractions and morphologies of the reaction products.

We show that the nitriding conditions can be manipulated to generate an AlN-reinforcement powder with control over the morphology and ratio of AlN and metallic Al. This provides a flexible and accessible method for the production of tailored AlN-reinforcement powders suited for powder metallurgy-based production of MMCs.

## 2. Materials and Methods

Aluminium (Al) powder was supplied by Chem-Supply and was irregular in shape with an average particle size of 50 μm. Magnesium (Mg) powder was supplied by Australia Metal Powders Supplies, was semi-spherical and had an average particle size of 100 μm. To promote the formation of AlN, trace amounts of Mg (1.0 wt.%) were mixed with the Al powder to serve as an oxygen getter and to disrupt the oxide layer on the surface of the Al powder [21,24,31].

The powder, consisting of 100 g blend of Al-1Mg, was weighed with an accuracy of ±0.0001 g using an A&D GR-200 scale and mixed in a three-dimensional tubular mixer for 60 min. For each nitriding experiment, approximately 3 g of the blended powder was placed on a stainless steel bed and heated in a controlled atmosphere horizontal tube furnace under a flowing nitrogen atmosphere.

Figure 1 shows the powder after mixing. Figure 1A shows the Al particles as supplied by Chem-Supply. The Al particles had an irregular morphology varying in size, with an average particle size of 50 μm. Figure 1B shows the Mg particles, which have an average particle size of 100 μm and a semi-spherical morphology. Figure 1C shows an EDS map of the blended powders pre-nitriding and shows the distribution of Mg particles within the Al powder mix.

The nitriding was achieved using a Carbolite STF 15/450 horizontal tube furnace equipped with a rotatory vacuum pump and high-purity nitrogen (Figure 2). The furnace chamber was purged to approximately 4.1 × 10${}^{-1}$ kPa to remove remnant oxygen by evacuating for 5 min followed by back-filling with nitrogen three times before heating. A high-purity nitrogen HP-N${}_{2}$ (>99.999%) atmosphere was used.

The samples were nitrided at temperatures between 490 and 580 ${}^{\circ }$C for 1, 3, 6 and 12 h. The nitrogen flow rate was maintained at 3 L/min for all experiments. All samples were heated in two stages. To reduce the risk of thermal shock to the ceramic furnace tube, it was initially pre-heated at 4 ${}^{\circ }$C/min to 50 ${}^{\circ }$C and held for 15 min, followed by a secondary heating stage of 3 ${}^{\circ }$C/min up to the nitriding temperature. The samples were cooled under flowing nitrogen.

The agglomerated products from the nitridation reaction were manually milled with a natural agate stone mortar and pestle to reprocess them into a powder. Subsequently, the powders were automatically sieved for 15 min through a series of mesh sizes (200 μm → 63 μm) to remove remaining agglomerations. A schematic showing the sequence of processing steps to nitride the loose Al powder to obtain the final Al/AlN-reinforcement powder is shown in Figure 3.

### Microstructure Characterisation

Phase analysis characterisation was conducted using a Bruker D8 advance powder X-ray diffraction (XRD) machine equipped with CuKα radiation at 40 kV and 40 mA. Data collection was conducted over a range of 20 to 90 (${}^{\circ }$2θ) with a scan time per step of 2${}^{\circ }$/min and increment step of 0.02${}^{\circ }$. Quantitative Phase Analysis (QPA) was performed using the powder diffraction software, PANalytical HighScore Plus [33,34,35,36]. A combination of internal standard and constant background quantification was applied to extract quantitative data from the XRD spectra. A nitrided sample (530 ${}^{\circ }$C for 6 h) was mixed with a known amount of the internal standard (10 wt.% of silicon). The results from the characterisation using the internal standard were used to establish a background used for the analysis of the other sample to enable accurate estimation of the phase proportions.

The microstructure and porosity were observed by optical microscopy (Reichert-Jung Polyvar Met Microscope) and captured using a digital camera (Canon EOS 5D Mark II). Furthermore, the samples were evaluated by scanning electron microscopy (SEM) using a Hitachi SU3500-A SEM at an accelerating voltage of 20 kV. In addition, the SEM was equipped with an Oxford X-max energy dispersive spectroscopy silicon drift detector (SDD EDS) to perform elemental analysis. Micro-hardness testing was conducted using a Struers Duramin DK2750 micro-hardness tester.

## 3. Results and Discussion

### 3.1. Nitridation Reaction

After the nitriding furnace treatment, the powder bed ranged from a light grey colour at lower temperatures (490 to 520 ${}^{\circ }$C) to dark black at higher temperatures (530 to 550 ${}^{\circ }$C) (Figure 4). This discolouration (pure AlN powder is white in appearance) is typical for the direct atmospheric nitridation of Al and may be due to the incorporation of oxygen and carbon impurities into the AlN structure [37]. At intermediate temperatures between 490 and 510 ${}^{\circ }$C, the darker agglomerated material was covered by a layer of loose light grey powder as shown in Figure 4C.

The loose powder, which was Al powder that had not reacted during the nitride treatment was removed before the subsequent analysis. For nitriding at ≤520 ${}^{\circ }$C, this comprised approximately 18 wt.% of the final total mass after nitriding. The proportion of discarded powder was dependent on the temperature. Nitriding at 530 ${}^{\circ }$C resulted in discarding roughly 2 wt.%, and, at temperatures ≥ 540 ${}^{\circ }$C, no loose powders were discarded.

The loose powders are believed to have been oxidised Al powder, which preferentially reacts with remnant oxygen in the furnace environment. A similar oxidised surface layer is also observed for Al-0.1 wt.%Mg powder compacts sintered under nitrogen atmospheres [26]. The Al powder acts as a self-getter removing remnant oxygen and reduces the oxygen particle pressure as the gas passes through the powder bed to the extent that nitridation occurs deeper in the powder bed [26].

The extent to which the agglomeration and consolidation of the powders occurred depended on both the temperature and duration of heat exposure (Figure 4). At lower temperatures (between 490 and 500 ${}^{\circ }$C for up to 6 h), the powders remained easily friable, and agglomerates were able to be broken apart and reprocessed back to a powder by hand using a mortar and pestle. For higher temperatures (≥510 ${}^{\circ }$C), the degree of agglomeration increased so that it became increasingly difficult to break the particles apart, and, at the highest temperatures (550 ${}^{\circ }$C), the powder bed was effectively consolidated into a single solid mass.

There are three mechanisms that potentially contribute to the agglomeration of the nitride powders. First, the sintering of the Al powders particles may occur aided by the low sublimation temperature of Mg (≈500 ${}^{\circ }$C) [38]. It is well known that the transfer of Mg vapour is a catalyst for rupture of the oxide on the surface of the Al, which may enable direct metallic contact between adjacent Al powder particles and subsequently facilitate solid-state sintering to occur [21].

Secondly, the exposure of underlying Al to the nitrogen atmosphere initiates an exothermic reaction to form AlN, which can lead to localised temperature increases. The exothermic heat releases associated with the formation of AlN and MgN are 317.98 and 288.70 kJ/mol, respectively, [39,40]. The large heat release associated with the exothermic reaction plays a significant role in the nitridation reaction. The reaction for the formation of AlN and MgN is represented by the chemical Formulas (Equation 1) and (Equation 2), respectively.
(1)$$2Al+N_{2}\to 2AlN\;\;\;\;\;\;\;\;\Delta H=317.98\;\left(\mathrm{kJ}/\mathrm{mol}\right)$$
(2)$$3Mg+N_{2}\to Mg_{3}N_{2}\;\;\;\;\;\;\;\;\Delta H=288.70\;\left(\mathrm{kJ}/\mathrm{mol}\right)$$

Several studies have identified exothermic heat release as a key factor for in situ nitridation processes, which may cause localised temperatures up to the order of 1700 ${}^{\circ }$C, sufficient to cause melting of Al and subsequently leading to agglomeration of the powders [41,42,43]. There is also a possibility for the localised formation of molten Al in the early stages of nitriding due to a eutectic reaction between the Al and Mg powder particles [23].

However, as detailed in the microstructural analysis below, no regions of localised Mg enrichment, which might be associated with the prior formation of a eutectic liquid, were detected within the reaction products. Finally, agglomeration after nitriding may be directly linked to the formation of the nitride layer and grows outward from the surface of the Al powders impinging on adjacent powders as it thickens, leading to interparticle bonding [1,10].

### 3.2. Morphology of Reaction Products

The influence of temperature on the morphology of the AlN formed from the Al powder is shown in the SEM micrograph in Figure 5. With increasing temperature, there are increasing amounts of nitride formed and coarsening of the AlN crystallites within the nitride layers. The microstructure transitions from fine nitride crystallites at temperatures ≤ 530 ${}^{\circ }$C to much coarser needles and whiskers at higher temperatures.

The nitride layer growth appears to have bi-directional growth characteristics, growing outward from the Al powder particle surface and into the Al powder interior. Similar observations were made for nitride layers produced by direct nitridation on Al plates under similar conditions, whereby there were two distinct layers that formed [23]. In that case, the outward-growing layer formed initially and evolved at a constant rate during nitridation, while the inward-growing layer formed later, and its growth rate accelerated with time.

Transmission electron microscopy (TEM) analyses on the nitride layers formed through these types of direct nitridation processes revealed that both layers were predominately comprised of micro- or nano-crystalline AlN comingled with small amounts of metallic Al [23]. From the micro-structures, it is evident that the outward-growing nitride layer can impinge on nitride layers formed on adjacent Al powder particles leading to interparticle bonding and agglomeration, such as in Figure 5B where we see two prior Al powder particles (dark contrast) surrounded by the accumulated nitride layer.

The flattened interface between these particles and their proximity suggests that solid-state sintering of the Al powders may precede the nitridation reaction. At high-temperatures around 550 ${}^{\circ }$C, the nitride is clearly coarsened and is comprised of long whiskers, featuring open void regions between where the metallic Al appears to have been consumed by the nitride whisker growth (Figure 5D).

At temperatures > 550 ${}^{\circ }$C, the microstructures indicate that, at the core of the powder bed remnant, Al was likely molten and formed a thin metal layer between the nitride Al regions. Figure 6 shows the thin Al band surrounding the nitride and enabling extensive agglomeration of the powder bed. Figure 6E shows that the thin bright band has a higher concentration of Al compared with the adjoining nitride regions.

Figure 7 shows an SEM micrograph and accompanying EDS maps of the nitride powders formed after nitriding at 510 ${}^{\circ }$C for 6 h. The maps display the elements Al and N. Only trace levels of Mg were detected within the nitride structures—insufficient for mapping. The limited detection of Mg supports assertions that Mg is sublimated during nitriding, and its limited detection in the reaction products may be due to removal under the nitrogen flow [10].

Figure 7A shows a polished cross-section of a partially nitride Al particle (light contrasting region) surrounded by the darker nitride layer. Figure 7B,C shows the EDS detection of N (blue) and Al (red) with increased levels of nitrogen within the nitride layer and predominantly Al detected within the core. This confirms that the outer region in Figure 7A is the AlN containing nitride layer, which has darker contrast due to the incorporation of the lighter N element with Al while the darker contrasting Al-rich region is remnant metallic Al.

A series of micro-hardness tests were conducted on the samples across the nitride layers and within the remnant metallic regions. The Al core of the particles showed hardness typical of that of pure Al (34.9 ± 7.3) Hv${}_{0.01}$, while the nitride regions exhibited hardness more than an order of magnitude greater (500.7 ± 63.2) Hv${}_{1.00}$. Hence, the nitride may be suitable as a reinforcement for the fabrication of MMCs, as they can impart considerable strengthening and wear resistance to the metal matrix.

### 3.3. Phase Analysis

Figure 8 shows the XRD spectra obtained from the powders and agglomerates after nitriding treatments at temperatures ranging from 490 to 550 ${}^{\circ }$C for a duration of 6 h. At 490 ${}^{\circ }$C, there was no significant AlN formed, and only the Al peaks were detected. However, AlN diffraction peaks were observed at 500 ${}^{\circ }$C and were present with increasing intensity as the temperature increased while the intensity of the Al peaks was reduced. Ostensibly, the increased intensity of AlN and reduced intensity of Al peaks with increasing temperature can be ascribed to the increased amounts of AlN formed by the consumption of metallic Al during nitridation. AlN is the predominant phase in all spectra for Al powders nitrided at temperatures ≥ 510 ${}^{\circ }$C.

Quantitative phase analysis was conducted on the XRD spectra to estimate the relative phase proportions after nitriding, and these are presented in Figure 9. For a 6 h nitridation period, there were no AlN peaks detected at 490 ${}^{\circ }$C. The critical temperature to activate nitridation was 500 ${}^{\circ }$C, which led to more than 40% AlN after 6 h with the bulk being metallic Al. After 6 h at 510 ${}^{\circ }$C, the proportion of AlN increased to above 70%, while, for temperatures ≥ 520 ${}^{\circ }$C and ≤540 ${}^{\circ }$C, the proportion of AlN was between 80% and 90%.

The maximum yield of AlN was 87% at 530 ${}^{\circ }$C, while there was a reduction to around 78% at the highest nitriding temperature (550 ${}^{\circ }$C). Of note, it appears that, once nitriding is activated, the formation of AlN occurs readily. This is evident in Figure 9B where more than 80% AlN is already present after 1 h of nitridation at 530 ${}^{\circ }$C. The exothermic heat release drives the reaction to proceed rapidly once initiated [4].

The formation of AlN by direct nitridation through similar methods has been reported to occur at temperatures down to 550 ${}^{\circ }$C [29,44,45]. Here, the QPA analysis results show that, under the current conditions, a high yield of AlN was obtained for nitriding temperatures down to 500 ${}^{\circ }$C. This offers a novel method for the tailored nitridation of AlN powder at significantly lower temperatures than comparable methods, which typically require temperatures of ≥ 600 ${}^{\circ }$C [4,46].

### 3.4. Synthesis of Aln Powder

Successful nitridation of the Al powder occurred for all temperatures between 500 and 560 ${}^{\circ }$C. However, lower nitriding temperatures ≤ 510 ${}^{\circ }$C were necessary to reduce the extent of agglomeration to retain friable AlN absent of large and difficult-to-break-up agglomerates, which could be readily processed into powder through the manual methods used in the study. The ability to readily process the reinforcement into powder is necessary for subsequent use as a reinforcement for the powder metallurgy fabrication of Al/AlN MMCs.

For 6 h nitriding, the AlN yields at 500 and 510 ${}^{\circ }$C were 42.60% and 73.83%, respectively. Figure 10 shows the nitrided and processed AlN powder (530 ${}^{\circ }$C for 6 h) produced by this method with uniform levels of Al and N detected by EDS (Figure 10C,D). The particle sizes range from 1 to 50 μm with an average particle size of 20 μm. The AlN powder is of low sphericity and has sub-angular morphologies.

Low-temperature direct nitriding enables a high degree of control over the AlN yields and morphology. Conventional powder metallurgy preparation of ex situ reinforced Al/AlN MMCs using commercial AlN is compromised by the inert nature of the reinforcement, which does not actively participate in the sintering, thereby, leading to restricted densification and poor interfacial bonding between the reinforcement and the Al matrix.

Here, the remnant metallic Al in the synthesised reinforcement may participate in solid-state sintering processes for powder metallurgy preparation of Al/AlN MMCs leading to enhanced densification and better bonding between the matrix and reinforcement than with conventional AlN reinforcements. The process for low-temperature direct nitridation of Al powders established in this research offers an economical means for the preparation of an AlN-reinforcement powder, which is suited to powder metallurgy preparation of Al/AlN MMCs.

## 4. Conclusions

We conducted a systematic study into the effects of temperature and time on the low-temperature direct nitriding of loose Al powder mixed with a 1 wt.% Mg atmosphere under high-purity nitrogen. The aim was to establish an effective regimen to prepare AlN-reinforcement powder suitable for powder metallurgy preparation of MMCs. Our experimental results support the following conclusions:Effective nitriding of the Al-1Mg powder mixture, verified by XRD spectra and SEM-EDS mapping, was obtained upon exposure to flowing high-purity nitrogen at temperatures ≤ 500 ${}^{\circ }$C.Extensive agglomeration of the reaction products occurred at higher temperatures (520 ${}^{\circ }$C and above). The extent of agglomeration increased with increasing nitriding temperature and with the duration of heating. Heat release associated with exothermic formation of AlN contributed to agglomeration. Nitriding at temperatures ≤ 510 ${}^{\circ }$C reduced agglomeration and retained a friable product, which was readily reprocessed into powder.The maximum AlN yield was 85% obtained at 530 ${}^{\circ }$C for times ≥ 6 h. However, a relatively high proportion of AlN could be attained (73% after 6 h at 510 ${}^{\circ }$C) while retaining excellent friability so that it could be manually reprocessed to powder.The synthesised reinforcement powder consisted of micro- and nano-crystalline AlN with metallic Al. The ratio of AlN to Al was able to be varied from 40% to 80% by adjusting the nitriding temperature between 500 and 520 ${}^{\circ }$C for heating times of 6 h. This provides an economic and flexible means for the preparation of tailored AlN-reinforcement powders suited to powder metallurgy-based preparation of MMCs.

## Figures and Tables

**Figure 1 materials-16-01583-f001:**
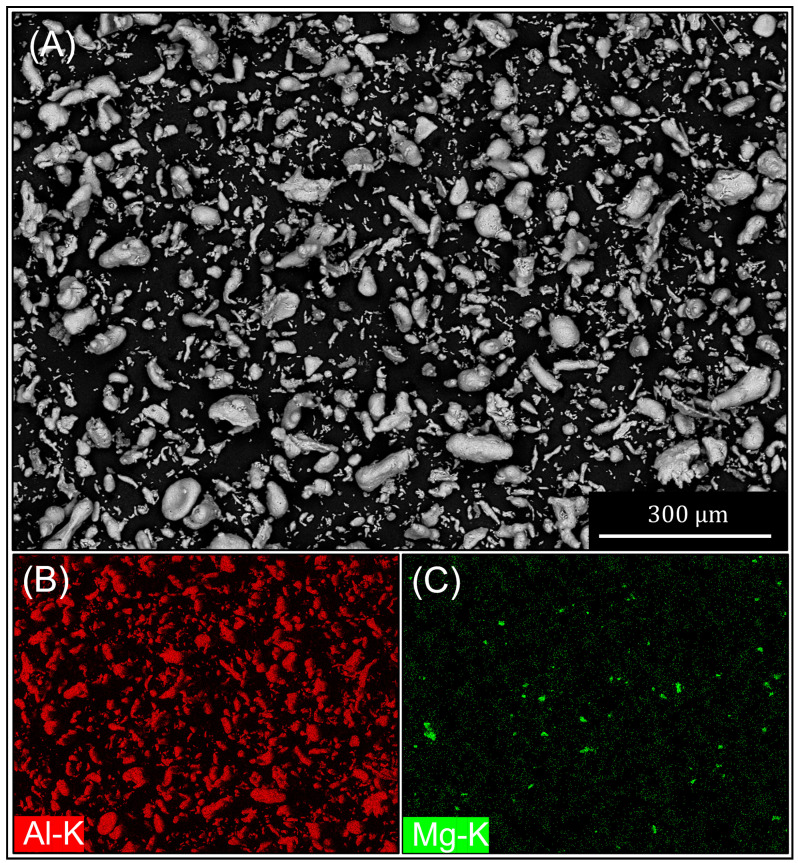
SEM micrograph and EDS mapping of the initial powder blend pre-nitriding: (**A**) SEM micrograph of Al-1Mg powder blend, (**B**) Al SEM-EDS map of the powder blend and (**C**) Mg SEM-EDS map of the powder blend showing the homogeneous distribution of Mg.

**Figure 2 materials-16-01583-f002:**
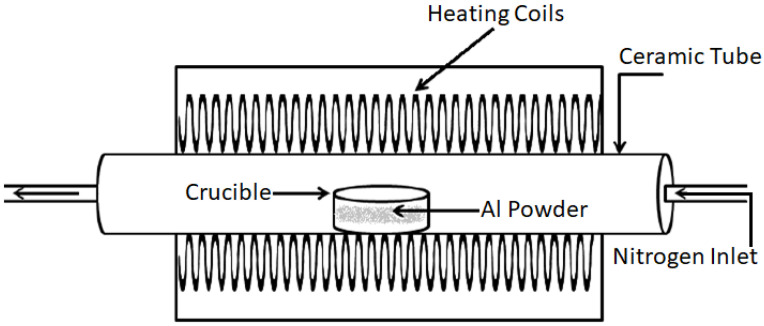
Schematic showing the setup for the low-temperature nitriding of Al + 1 wt.% Mg powder using a horizontal tube furnace equipped with a high-purity nitrogen feed [32].

**Figure 3 materials-16-01583-f003:**
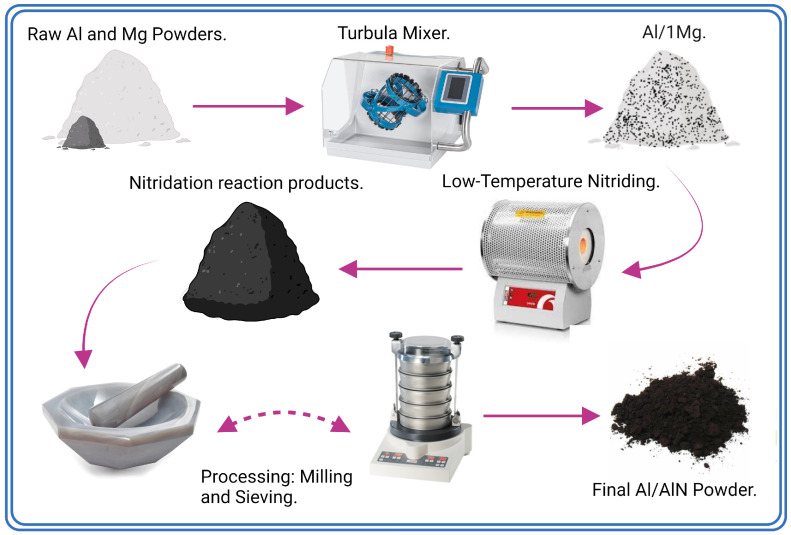
Schematic flowchart showing the processing steps used to nitride the loose Al powder to obtain the final nitride powders: Al/Mg powder blending, low-temperature nitriding and processing via mortar and pestle, (iterative to achieve ensure homogeneity in particle size, ≤63 μm).

**Figure 4 materials-16-01583-f004:**
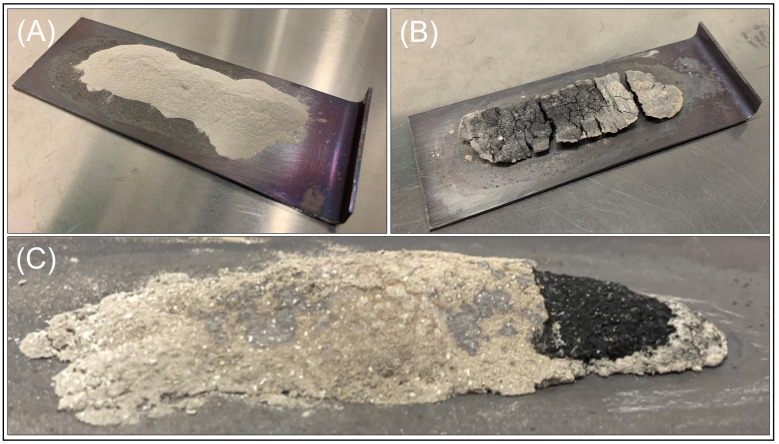
Images of the powder bed pre- and post-nitriding. (**A**) Pure Al powder pre-nitriding, (**B**) Al powder nitrided at 530 ${}^{\circ }$C for 6 h resulting in fully agglomerated Al:AlN and (**C**) Al powder nitrided for 6 h at 510 ${}^{\circ }$C resulting in partially agglomerated Al/AlN, which was sufficiently friable to manually process into a powder.

**Figure 5 materials-16-01583-f005:**
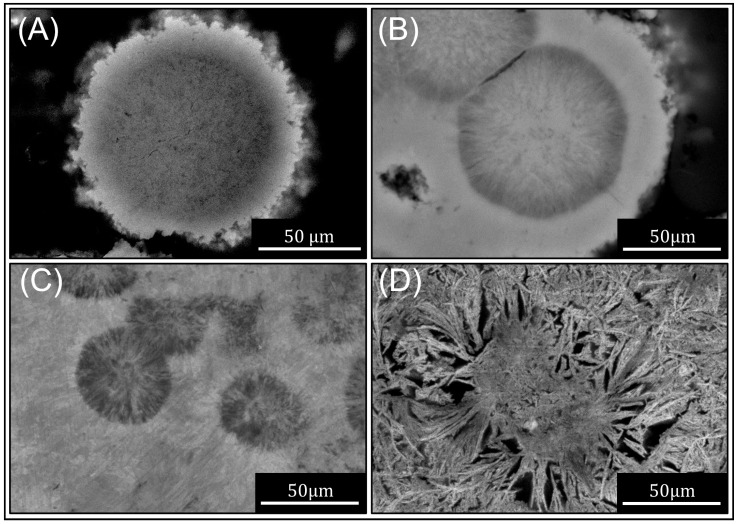
SEM micrograph of nitrided Al powder that has been mounted and polished. Temperatures ranging from 500 to 550 ${}^{\circ }$C: 6 h at (**A**) 500 ${}^{\circ }$C, (**B**) 510 ${}^{\circ }$C, (**C**) 530 ${}^{\circ }$C and (**D**) 550 ${}^{\circ }$C.

**Figure 6 materials-16-01583-f006:**
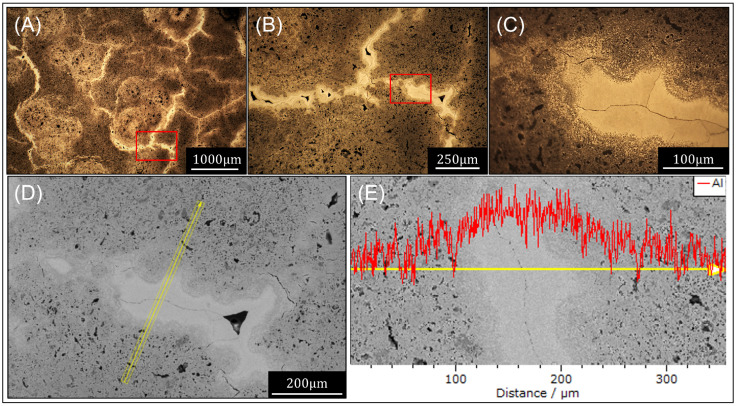
Effects of nitriding at temperatures ≥ 550 ${}^{\circ }$C: (**A**–**C**) optical image showing bright thin metallic bands dispersed throughout the agglomerated AlN sample and (**D**–**E**) SEM-EDS line profile scan across a metallic band showing the high detection of Al.

**Figure 7 materials-16-01583-f007:**
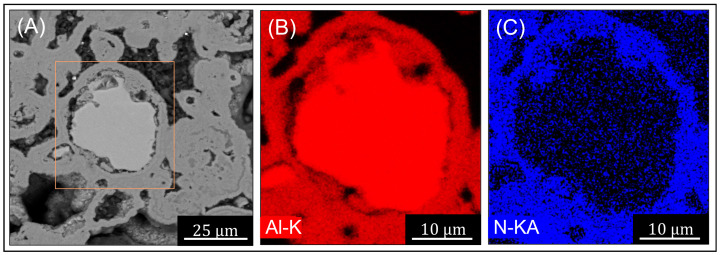
SEM micrograph and EDS maps of powder after nitriding at 510 ${}^{\circ }$C for 6 h: (**A**) SEM micrograph of nitride Al particle, (**B**) EDS map Al-K Map and (**C**) EDS map N-KA.

**Figure 8 materials-16-01583-f008:**
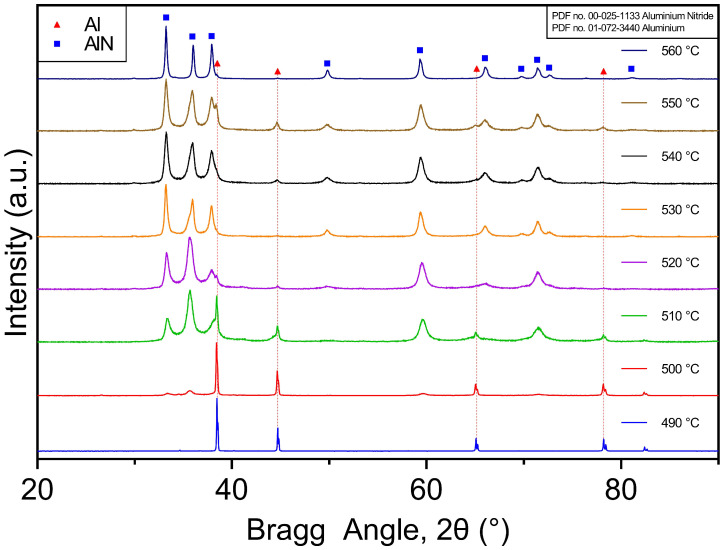
X-ray diffraction spectra from Al powder samples exposed to nitrogen atmospheres for 6 h at temperatures between 490 and 560 ${}^{\circ }$C showing metallic Al peaks and an increasing intensity of peaks associated with AlN with increasing nitridation temperature (PDF no. 01-072-3440 Aluminium, PDF no. 00-025-1133 Aluminium Nitride).

**Figure 9 materials-16-01583-f009:**
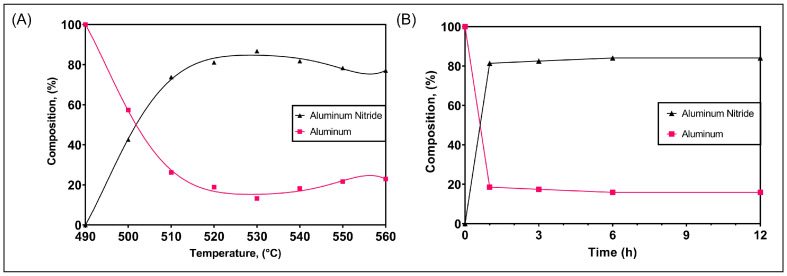
Proportions of AlN formed with temperature and time: (**A**) quantitative phase analysis results of the proportions of Al and AlN after nitriding for 6 h at various temperatures and (**B**) quantitative phase analysis of the proportions of Al and AlN after nitriding at 530 ${}^{\circ }$C for 1, 3, 6 and 12 h.

**Figure 10 materials-16-01583-f010:**
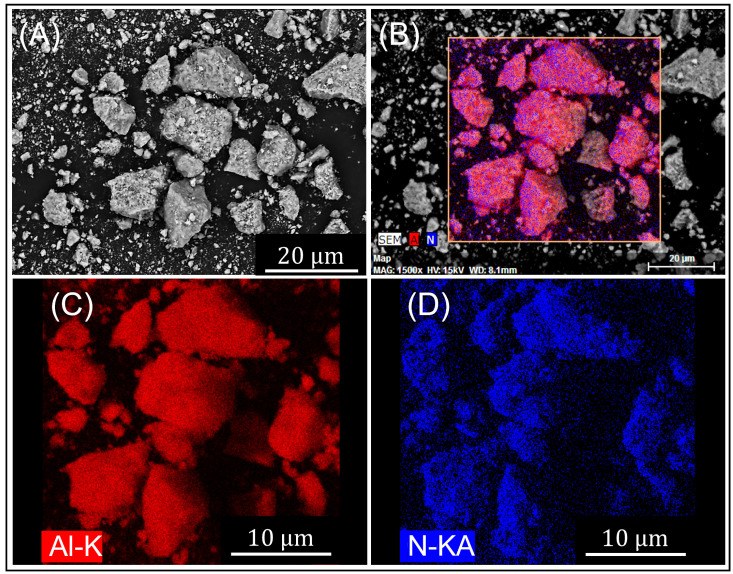
SEM micrograph and EDS mapping of final AlN containing powder nitrided at 520 ${}^{\circ }$C for 6 h, hand milled and sieved at 63 μm: (**A**) SEM image of processed powder, (**B**) EDS map scan region detecting Al and N, (**C**) EDS map Al-K and (**D**) EDS map N-KA.

## Data Availability

All data in this paper were obtained from our experiment and are authentic and reliable. The publication of the data obtained the consent of all authors.

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
