# Peer review of "Impacts of Temperature and Time on Direct Nitridation of Aluminium Powders for Preparation of AlN Reinforcement"

_materials, 2023, doi:10.3390/ma16041583_

Round 1
Reviewer 1 Report
I have had the opportunity to review the article " Impacts of temperature and time on direct nitridation of aluminium powders". In order to get the full potential of the paper, some minor corrections need to be done. Some comments are provided below:
What is new being presented in this article? Put this information in the abstract. Abstract section should be concisely reflected the content and summarize the problem, the method, the results, and the conclusions. The abstract needs to be improved. Demonstrate in the abstract novelty, practical significance. Also, please add more qualitative and quantitative results of your work.
Wherever applicable, the scientific explanation needs to be added and the research novelties need to be clearly emphasized.
The gap area in the research is not clear.
At the end of Introduction section, it would be better to add the paper's organization in different sections.
Some giving citations need to be check such as [1-4] [30-33], as they may not provide the required information in a sentence. Besides, add scientific novelty and practical relevance. Add a clear purpose to the article. Please show the literature gaps demonstrating the presented study fills it. At the last paragraph of the introduction, please clearly show the general outline of the paper and show the importance of the study along with the main aim.
The authors should add more information about the experimental method and the test equipment specification.
Further, results and analysis of experiments should be compared with previous researchers by citing references
Improve the conclusion. It is necessary to more clearly show the novelty of the article and the advantages of the proposed method. What is the difference from previous work in this area? Show practical relevance. The article is interesting, but needs to be improved. Authors should carefully study the comments and make improvements to the article step by step. Add 4-5 items of the findings of the study.
The XRD diffraction pattern of Fig.7 should be marked with a standard PDF card. Also, the referred PDF card No. in XRD results should be supplemented in the corresponding text of Fig. 7.
Improve the sharpness of all low quality images and their scale and numbers especially for Figs. 1, 4, 5, 7.
Please check the manuscript for wrong choice of words, grammatical errors and incoherent sentence structure. The authors can use suitable grammar-checking software / use the help of a native English speaker to correct these mistakes. Please fix the typographical and eventual language problems in paper.
Reviewer 2 Report
...
The manuscript entitled "Impacts of temperature and time on direct nitridation of aluminum powders" by Samuel Rogers, Matthew S. Dargusch, Damon Kent is devoted to describing the original method for the synthesis of aluminum nitride and determining its optimal conditions. The subject of the manuscript corresponds to the objectives of the publication Materials. The manuscript describes in detail the conditions of the experiment, its results, and possible mechanisms. In general, the manuscript left a very good impression and can be recommended for publication after removing some of the comments.
115 "mortar and pestle" Specify the material of the mortar and pestle.
167 "exothermic reaction to form AlN" Give specific thermodynamic data for this reaction.
Figure 4. "(A) 6 h at 500 °C, (B) 6 h at 510 °C, (C) 6h at 530 °C, and (D) 6 h at 550" -- > " 6 h at (A) 500 °C, (B) 510 °C, (C) 530 °C, and (D) 550"
202-203 "Only the elements Al and N were detected and are shown." That is, magnesium was not detected at all? Doesn't this seem strange to the authors, given that magnesium has a melting point of 650°C?
Figure 8. Indicate the error bars in the figure. Expand the range of the temperature and time axes for greater clarity of the figure. Try to remove the black stroke of the charts.
247-248 "Particle size ranges from 1 µm to 50 µm with an average particle size of 20 µm." Please provide a particle size distribution graph.
269 "melting of metallic Al" The melting point of aluminum is much higher than 550°C. Did the authors consider the possibility of local formation between aluminum and magnesium of a eutectic composition with a low melting point (for example, Al3Mg2 - 452оС - https://www.mdpi.com/materials/materials-12-02661/article_deploy/html/images/materials-12-02661-g001-550.jpg ), which may have been the cause of the observed phenomena.
...
Reviewer 3 Report
Reveree report on “Impacts of temperature and time on direct nitridation of aluminium powders”
In general, this is good article that can be recommended for publication after the authors will provide a proper answer for a few comments/questions that have arisen during a detailed reading of the work.
1. The authors did not sufficiently reveal the relevance and novelty of the work. Most of the literature references are quite old and it is not clear whether this problem is still of interest or the authors are not up to date with the latest work.
2. In the manuscript, the following problem is completely silent the properties of the macroporous AlN depend on point defects and defects on the surface, which were comprehensively studied both theor and experimental and documented in many works on bulk and nanomaterials. Although such defects can be created in different ways, (including irradiation), their properties are almost the same. See, fore example:
Bellucci, S., et al (2007). Luminescence, vibrational and XANES studies of AlN nanomaterials. Radiation measurements, 42(4-5), 708-711.
Weinstein, I. A., Vokhmintsev, A. S., & Spiridonov, D. M. (2012). Thermoluminescence kinetics of oxygen-related centers in AlN single crystals. Diamond and related materials, 25, 59-62.
Zhukovskii, Y. F., et al (2007). Influence of F centres on structural and electronic properties of AlN single-walled nanotubes. Journal of Physics: Condensed Matter, 19(39), 395021.
Therefore, the obvious question arises how defective the resulting samples are and how it depends on the original material.
3. Figure 4. Its quality is not enough.
4. Figure 5 D. Its quality is also not good.
Round 2
Reviewer 3 Report
After a sufficiently constructive revision, the manuscript can be accepted.